# SMOOTH MIN-MAX MONOTONIC NETWORKS

## ABSTRACT

Monotonicity constraints are powerful regularizers in statistical modelling. They can support fairness in computer-aided decision making and increase plausibility in data-driven scientific models. The seminal min-max (MM) neural network architecture ensures monotonicity, but often gets stuck in undesired local optima during training because of partial derivatives being zero when computing extrema. We propose a simple modification of the MM network using strictly-increasing smooth minimum and maximum functions that alleviates this problem. The resulting smooth min-max (SMM) network module inherits the asymptotic approximation properties from the MM architecture. It can be used within larger deep learning systems trained end-to-end. The SMM module is conceptually simple and computationally less demanding than state-of-the-art neural networks for monotonic modelling. Our experiments show that this does not come with a loss in generalization performance compared to alternative neural and non-neural approaches.

## 1 INTRODUCTION

In many data-driven modelling tasks we have a priori knowledge that the output is monotonic, that is, non-increasing or non-decreasing, in some of the input variables. This knowledge can act as a regularizer, and often monotonicity is a strict constraint for ensuring the plausibility and therefore acceptance of the resulting model. We are particularly interested in monotonicity constraints when learning bio- and geophysical models from noisy observations, see Figure 1. Examples from finance, medicine and engineering are given, for instance, by Daniels & Velikova, 2010, see also the review by Cano et al. (2019). Monotonicity constraints can incorporate ethical principles into data-driven models and improve their fairness (e.g., see Cole & Williamson, 2019; Wang & Gupta, 2020).

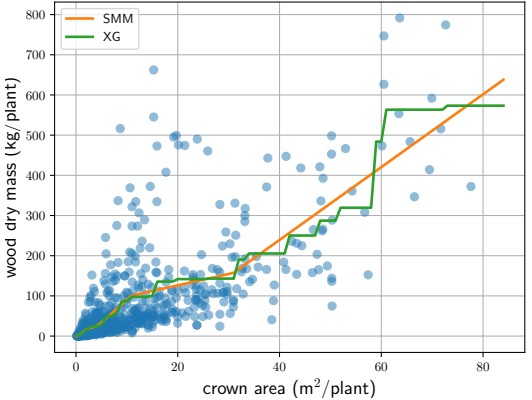

Figure 1: Learning an allometric equation from data with XGBoost (XG) and a smooth min-max network (SMM), here estimating wood dry mass (and thereby stored carbon) from tree crown area (Hiernaux et al., 2023; Tucker et al., 2023).

Work on monotonic neural networks was pioneered by the min-max (MM) architecture proposed by Sill (1997), which is simple, elegant, and able to asymptotically approximate any monotone target function by a piecewise linear neural network model. However, learning an MM network, which can be done by unconstrained gradient-based optimization, often does not lead to satisfactory results. Thus, a variety of alternative approaches were proposed, which are much more complex than an MM network module (for recent examples see Milani Fard et al., 2016; You et al., 2017; Gupta et al., 2019; Yanagisawa et al., 2022; Sivaraman et al., 2020; Liu et al., 2020, and Nolte et al., 2022). We argue that the main problem when training an MM network are partial derivatives being zero because of the maximum and minimum computations. This leads to large parts of the MM network being silent, that is, most parameters of the network do not contribute to computing the model output at all, and therefore the MM network underfits the training data with a very coarse piecewise linear approximation. We alleviate this issue by replacing the maximum and minimum by

smooth *and monotone* counterparts. The resulting neural network module is referred to as *smooth min-max* (SMM) and exhibits the following properties:

- The SMM network inherits the asymptotic approximation properties of the min-max architecture, but does not suffer from large parts of the network not being used after training.

- The SMM module can be used within a larger deep learning system and be trained end-to-end using unconstrained gradient-based optimization in contrast to standard isotonic regression and (boosted) decision trees.

- The SMM module is simple and does not suffer from the curse of dimensionality when the number of constrained inputs increases, in contrast to lattice based approaches.

- The function learned by SMM networks is smooth in contrast to isotonic regression, linearly interpolating lattices, and boosted decision trees.

- Our experiments show that the advantages of SMM do not come with a loss in performance. In experiments on elementary target functions, SMM compared favorably with min-max networks, isotonic regression, XGBoost, expressive Lipschitz monotonic networks, and hierarchical lattice layers; and SMM also worked well on partial monotone real-world benchmark problems.

We would like to stress that the smoothness property is not just a technical detail. It influences how training data are inter- and extrapolated, and smoothness can be important for scientific plausibility. Figure 1 shows an example where an allometric equation is learned from noisy observations using the powerful XGBoost (Chen & Guestrin, 2016) and a simple SMM layer. In this example, the output (wood dry mass) should be continuously increasing with the input (tree crown area). While both machine learning models give good fits in terms of mean squared error, neither the staircase shape nor the constant extrapolation of the tree-based model are scientifically plausible.

The next section will present basic theoretical results on neural networks with positive weights and the MM architecture as well as a brief overview of interesting alternative neural and non-neural approaches to monotonic modelling. After that, Section 3 will introduce the SMM module and show that it inherits the asymptotic approximation properties from MM networks. Section 4 will present an empirical evaluation of the SMM module with a clear focus on the monotonic modelling capabilities in comparison to alternative neural and non-neural approaches before we conclude in Section 5.

## 2 BACKGROUND

A function $f(\boldsymbol{x})$ depending on $\boldsymbol{x} = (x_1, \ldots, x_d)^{\mathrm{T}} \in \mathbb{R}^d$ is non-decreasing in variable $x_i$ if $x_i' \geq x_i$ implies $f(x_1, \ldots, x_{i-1}, x_i', x_{i+1}, \ldots, x_d) \geq f(x_1, \ldots, x_{i-1}, x_i, x_{i+1}, \ldots, x_d)$; being non-increasing is defined accordingly. A function is called monotonic if it is non-increasing or non-decreasing in all $d$ variables. Without loss of generality, we assume that monotonic functions are non-decreasing in all $d$ variables (if the function is supposed to be non-increasing in a variable $x_i$ we simply negate the variable and consider $-x_i$). We address the task of inferring a monotonous model from noisy measurements. For regression we are given samples $\mathcal{D}_{\mathrm{train}} = \{(\boldsymbol{x}_1, y_1), \ldots, (\boldsymbol{x}_n, y_n)\}$ where $y_i = f(\boldsymbol{x}_i) + \varepsilon_i$ with $f$ being monotonic and $\varepsilon_i$ being a realization of a random variable with zero mean. Because of the random noise, $\mathcal{D}_{\mathrm{train}}$ is not necessarily a monotonic data set, which implies that interpolation does in general not solve the task.

### 2.1 NEURAL NETWORKS WITH POSITIVE WEIGHTS

**Basic theoretical results.** A common way to enforce monotonicity of canonical neural networks is to restrict the weights to be non-negative. If the activation functions are monotonic, then a network with non-negative weights is also monotonic (Sill, 1997; Daniels & Velikova, 2010). However, this does not ensure that the resulting network class can approximate any monotonous function arbitrarily well. If the activation functions of the hidden neurons are standard sigmoids (logistic/Fermi functions) and the output neuron is linear (e.g., the activation function is the identity), then a neural network with positive weights and at most $d$ layers can approximate any continuous function mapping from a compact subset of $\mathbb{R}^d$ to $\mathbb{R}$ arbitrarily well (Daniels & Velikova, 2010, Theorem 3.1). Interesting recent theoretical work by Mikulincer & Reichman (2022) shows that with Heaviside step activation functions the above result can be achieved with four layers for non-negative inputs. However, if the

activation functions in the hidden layers are convex, such as the popular (leaky) ReLU and ELU (Nair & Hinton, 2010; Maas et al., 2013; Clevert et al., 2016) activation functions, then a canonical neural network with positive weights is a combination of convex functions and as such convex, and accordingly one can find a non-convex monotonic function that cannot be approximated within an a priori fixed additive error (Mikulincer & Reichman, 2022, Lemma 1). However, their interpolation results assume monotone data and are therefore not applicable to the general case of noisy data.

**Min-max networks.** Min-max (MM) networks as proposed by Sill (1997) are a concave combination – taking the minimum – of convex combinations – taking the maximum – of monotone linear functions, where the monotonicity is ensured by positive weights, see Figure 2. The architecture comprises $K$ groups of linear neurons, where, following the original notation, the number of neurons in group $k$ is denoted by $h_k$. Given an input $\boldsymbol{x} \in \mathbb{R}^d$, neuron $j$ in group $k$ computes

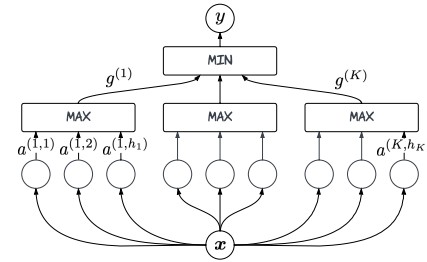

$$a^{(k,j)}(\boldsymbol{x}) = \boldsymbol{w}^{(k,j)} \cdot \boldsymbol{x} - b^{(k,j)} \tag{1}$$

with weights $\boldsymbol{w}^{(k,j)} \in (\mathbb{R}_0^+)^d$ and bias $b^{(k,j)} \in \mathbb{R}$. Then all $h_k$ outputs within a group $k$ are combined via

Figure 2: Schema of a min-max module.

$$g^{(k)}(\boldsymbol{x}) = \max_{1 \leq j \leq h_k} a^{(k,j)}(\boldsymbol{x}) \tag{2}$$

and the output of the network is given by

$$y(\boldsymbol{x}) = \min_{1 \leq k \leq K} g^{(k)}(\boldsymbol{x}) \ . \tag{3}$$

For classification tasks, $y$ can be interpreted as the logit. To ensure positivity of weights during unconstrained optimization, we encode each weight $w_i^{(k,j)}$ by an unconstrained parameter $z_i^{(k,j)}$, where $w_i^{(k,j)} = \exp\left(z_i^{(k,j)}\right)$ (Sill, 1997) or $w_i^{(k,j)}$ results from squaring (Daniels & Velikova, 2010) or applying the exponential linear function (Cole & Williamson, 2019) to $z_i^{(k,j)}$. The order of the minimum and maximum computations can be reversed (Daniels & Velikova, 2010). The convex combination of concave functions gives the following asymptotic approximation capability:

**Theorem 1** (Sill, 1997; Daniels & Velikova, 2010). *Let $f(\boldsymbol{x})$ be any continuous, bounded monotonic function with bounded partial derivatives, mapping $[0, 1]^D$ to $\mathbb{R}$. Then there exists a function $f_{\text{net}}(\boldsymbol{x})$ which can be implemented by a monotonic network such that $|f(\boldsymbol{x}) - f_{\text{net}}(\boldsymbol{x})| < \epsilon$ for any $\epsilon > 0$ and any $\boldsymbol{x} \in [0, 1]^d$.*

## 2.2 RELATED WORK

**Lattice layers.** Neural networks with lattice layers constitute a state-of-the-art approach for incorporating monotonicity constraints (Milani Fard et al., 2016; You et al., 2017; Gupta et al., 2019; Yanagisawa et al., 2022). A lattice layer defines a hypercube with $L^d$ vertices. The integer hyperparameter $L > 1$ defines the granularity of the hypercube and $d$ is the input dimensionality, which is replaced by the number of input features with monotonicity constraints in *hierarchical lattice layers* (HLLs, Yanagisawa et al., 2022). In contast to the original lattice approaches, a HLL can be trained by unconstrained gradient-based-optimization. The $L^d$ scaling of the number of parameters is a limiting factor. For larger $d$, the task has to be broken down using an ensemble of several lattice layers, each handling fewer constraints Milani Fard et al. (2016)

**Certified monotonic neural networks.** A computationally very expensive approach to monotonic modelling is to train standard piece-wise linear (ReLU) networks and to ensure monotonicity afterwards. Liu et al. (2020) propose to train with heuristic regularization that favours monotonicity. After training, it is checked by solving a MILP (mixed integer linear program) if the network fulfills all constraints. If not, the training is repeated with stronger regularization. Sivaraman et al. (2020) suggest to adjust the output of the trained network to ensure monotonicity. This requires solving an SMT (satisfiability modulo theories, a generalization of SAT) problem for each prediction.

**Lipschitz monotonic networks.** The approach closest to our work are Lipschitz monotonic networks (LMNs) recently proposed by Nolte et al. (2022). The idea of LMNs is to ensure that a base model is $\lambda$-Lipschitz with respect to the $L^1$-norm and then to add $\lambda x_i$ to the model for each constrained input $i$. LMNs are smooth and can be trained end-to-end. The LMN approach requires choosing the Lipschitz constant $\lambda$. To enforce the Lipschitz property of neural models, normalization of the weight matrices is added. However, to ensure that the networks can approximate any monotonic Lipschitz bounded function, one has to additionally use special activation functions to prevent "gradient attenuation" (in the experiments by Nolte et al. the GroupSort activation function was used), see also Anil et al. (2019). This approximation result is slightly weaker than Theorem 1 in the sense that the choice of $\lambda$ constrains the class of functions the LMN can approximate.

**Non-neural approaches.** There are many approaches to monotonic prediction not based on neural networks, we refer to Cano et al. (2019) for a survey. We would like to highlight isotonic regression (Iso), which is often used for classifier calibration (e.g., see Niculescu-Mizil & Caruana, 2005). In its canonical form (e.g., see Best & Chakravarti, 1990 and De Leeuw et al., 2009), Iso fits a piecewise constant function to the data and is restricted to univariate problems. The popular XGBoost gradient boosting library (Chen & Guestrin, 2016) also supports monotonicity constraints. XGBoost incrementally learns an ensemble of decision trees; accordingly, the resulting regression function is piece-wise constant.

## 3 SMOOTH MONOTONIC NETWORKS

We now introduce the smooth min-max (SMM) network module, which addresses problems of the original MM architecture. The latter often performs worse than alternative approaches both in terms of training and test error, and the outcome of the training process strongly depends on the initialization. Even if an MM architecture has enough neurons to be able to approximate the underlying target functions well (see Theorem 1), the neural network parameters realizing this approximation may not be found by the (gradient-based) learning process. When using MM modules in practice, they often underfit the training data and seem to approximate the data using a piecewise linear model with very few pieces — much less than the number of neurons. This observation is empirically studied in Section 4.1. We say that neuron $j^*$ in group $k^*$ in an MM unit is *active* for an input $\boldsymbol{x}$, if $k^* = \arg\min_{1 \le k \le K} g^{(k)}(\boldsymbol{x})$ and $j^* = \max_{1 \le j \le h_k} a^{(k,j)}(\boldsymbol{x})$. A neuron is *silent* over a set of inputs $\mathcal{X} \subset \mathbb{R}^d$ if it is not active for any $\boldsymbol{x} \in \mathcal{X}$. If neuron $j$ in group $k$ is silent over all inputs from some training set $\mathcal{D}_{\text{train}}$, we have $\partial y / \partial a^{(k,j)}(\boldsymbol{x}) = 0$ for all $\boldsymbol{x} \in \mathcal{X}$. Once a neuron is silent over the training data, which can easily be the case directly after initialization or happen during training, there is a high chance that gradient-based training will not lead to the neuron becoming active. Indeed, our experiments in Section 4.1 show that only a small fraction of the neurons in an MM module are active when the trained model is evaluated on test data.

The problem of silent neurons and the lack of smoothness can be addressed by replacing the minimum and maximum operation in the MM architecture by smooth counterparts. Not every smooth approximation to the maximum/minimum function is suitable, it has to preserve monotonicity. A strictly increasing approximation to the maximum is the LogSumExp function. Let $x_1, \ldots, x_n \in \mathbb{R}$. We define the scaled LogSumExp function with scaling parameter $\beta > 0$ as

$$\mathrm{LSE}_\beta(x_1, \ldots, x_n) = \frac{1}{\beta} \log \sum_{i=1}^n \exp(\beta x_i) = \frac{1}{\beta} \left( c + \log \sum_{i=1}^n \exp(\beta x_i - c) \right) \;, \qquad (4)$$

where the constant $c$ can be freely chosen to increase numerical stability, in particular as $c = \max_{1 \le i \le n} x_i$. The functions $\mathrm{LSE}_\beta(\mathcal{X})$ and $\mathrm{LSE}_{-\beta}(\mathcal{X})$ are smooth and monotone increasing in $x_1, \ldots, x_n$. It holds:

$$\max_{1 \le i \le n} x_i < \mathrm{LSE}_\beta(x_1, \ldots, x_n) \le \max_{1 \le i \le n} x_i + \frac{1}{\beta} \ln(n) \qquad (5)$$

$$\min_{1 \le i \le n} x_i - \frac{1}{\beta} \ln(n) \le \mathrm{LSE}_{-\beta}(x_1, \ldots, x_n) < \min_{1 \le i \le n} x_i \qquad (6)$$

The proposed SMM module is identical to an MM module, except that eqs. 2 and 3 are replaced by

$$g_{\text{SMM}}^{(k)}(\boldsymbol{x}) = \text{LSE}_\beta \left( a^{(k,1)}(\boldsymbol{x}), \ldots, a^{(k,h_k)}(\boldsymbol{x}) \right) \text{ and} \tag{7}$$

$$y_{\text{SMM}}(\boldsymbol{x}) = \text{LSE}_{-\beta} \left( g_{\text{SMM}}^{(1)}(\boldsymbol{x}), \ldots, g_{\text{SMM}}^{(K)}(\boldsymbol{x}) \right) \ . \tag{8}$$

We treat $\beta$, properly encoded to ensure positvity, as an additional learnable parameter. Each scaled LogSumExp operation could have its own learnable $\beta$ parameter, however, we did not find this necessary in our experiments. Thus, the number of parameters of an SMM module is $1 + (d + 1) \sum_{k=1}^{K} h_k$. If the target function is known to be (strictly) concave, we can set $K = 1$ and $h_1 > 1$; if it is known to be convex, we set $K > 1$ and can set $h_k = 1$ for all $k$. The default choice is $K = h_1 = h_2 = \cdots = h_K$.

**Approximation properties.** The SMM inherits the approximation properties from the MM, e.g.:

**Corollary 1.** *Let $f(\boldsymbol{x})$ be any continuous, bounded monotonic function with bounded partial derivatives, mapping $[0, 1]^D$ to $\mathbb{R}$. Then there exists a function $f_{\text{smooth}}(\boldsymbol{x})$ which can be implemented by a smooth monotonic network such that $|f(\boldsymbol{x}) - f_{\text{smooth}}(\boldsymbol{x})| < \epsilon$ for any $\epsilon > 0$ and any $\boldsymbol{x} \in [0, 1]^D$.*

*Proof.* Let $\epsilon = \gamma + \delta$ with $\gamma > 0$ and $\delta > 0$. From Theorem 1 we know that there exists an MM network $f_{\text{net}}$ with $|f(\boldsymbol{x}) - f_{\text{net}}(\boldsymbol{x})| < \gamma$. Let $f_{\text{smooth}}$ be the smooth monotonic neural network with the same weights and bias parameters as $f_{\text{net}}$, in which the maximum and minimum operation have been replaced by $\text{LSE}_\beta$ and $\text{LSE}_{-\beta}$, respectively. Let $H = \max_{h=1}^{K} h_k$. For all $\boldsymbol{x}$ and groups $k$ we have $g_{\text{SMM}}^{(k)}(\boldsymbol{x}) = \text{LSE}_\beta \left( a^{(k,1)}(\boldsymbol{x}), \ldots, a^{(k,h_k)}(\boldsymbol{x}) \right) \leq \max_{1 \leq j \leq h_k} a^{(k,j)}(\boldsymbol{x}) + \frac{1}{\beta} \ln(h_k) \leq g^{(k)}(\boldsymbol{x}) + \frac{1}{\beta} \ln(H)$. Thus, also $y_{\text{SMM}}(\boldsymbol{x}) \leq y(\boldsymbol{x}) + \frac{1}{\beta} \ln(H)$. Similarly, we have $y_{\text{SMM}}(\boldsymbol{x}) = \text{LSE}_{-\beta} \left( g_{\text{SMM}}^{(1)}(\boldsymbol{x}), \ldots, g_{\text{SMM}}^{(K)}(\boldsymbol{x}) \right) \geq \text{LSE}_{-\beta} \left( g^{(1)}(\boldsymbol{x}), \ldots, g^{(K)}(\boldsymbol{x}) \right) \geq \min_{1 \leq k \leq K} g^{(k)}(\boldsymbol{x}) - \frac{1}{\beta} \ln(K) = y(\boldsymbol{x}) - \frac{1}{\beta} \ln(K)$. Thus, setting $\beta = \delta^{-1} \ln \max(K, H)$ ensures for all $\boldsymbol{x}$ that $|f_{\text{net}}(\boldsymbol{x}) - f_{\text{smooth}}(\boldsymbol{x})| \leq \delta$ and therefore $|f(\boldsymbol{x}) - f_{\text{smooth}}(\boldsymbol{x})| < \gamma + \delta = \epsilon$. $\qquad\square$

**Partial monotonic SMM.** Let $\mathcal{X}$ be a subset of variables from $\{x_1, \ldots, x_d\}$. Then a function is partial monotonic in $\mathcal{X}$ if it is monotonic in all $x_i \in \mathcal{X}$. The min-max and smooth-mini-max modules are partial monotonic in $\mathcal{X}$ if the positivity constraint is imposed for weights connecting to $x_i \in \mathcal{X}$ (Daniels & Velikova, 2010); the other weights can vary freely. However, more general module architectures are possible. Let us split the input vector into $(\boldsymbol{x}^c, \boldsymbol{x}^u)$, where $\boldsymbol{x}_c$ comprises all $\mathcal{X}$ and $\boldsymbol{x}_u$ the remaining $x_i \notin \mathcal{X}$. Let $\Psi^{(k,j)} : \mathbb{R}^{d-|\mathcal{X}|} \to [0, 1]^{|\mathcal{X}|}$ and $\Phi^{(k,j)} : \mathbb{R}^{d-|\mathcal{X}|} \to \mathbb{R}^{l^{(k,j)}}$ for some integer $l^{(k,j)}$ denote neural subnetworks for each neuron $i = 1, \ldots, h_k$ in each group $k = 1, \ldots, K$ (which may share weights). Then replacing equation 1 by $a^{(k,j)}(\boldsymbol{x}) = \boldsymbol{w}^{(k,j)} \cdot \boldsymbol{x} + \Psi^{(k,j)}(\boldsymbol{x}_u) \cdot \boldsymbol{x}_c + \boldsymbol{w}_u^{(k,j)} \cdot \Phi(\boldsymbol{x}_u) - b^{(k,j)}$ with $\boldsymbol{w}_u^{(k,j)} \in \mathbb{R}^{l^{(k,j)}}$ and $\forall m \in \mathcal{X} : w_m^{(k,j)} \geq 0$ preserves the constraints.

## 4 EXPERIMENTS

We empirically compared different monotonic modelling approaches on well-understood benchmark functions. We also present results for various partial monotonic real-world data sets. As in related studies, the results on the partial monotonic real-world data reflect the general inductive bias of the overall system architecture, not only the performance of the network modules handling monotonicity constraints; this bears the risk that the processing of the unconstrained features occludes the monotonic modelling performance.

In our experiments, we assumed that we do not have any prior knowledge about the shape of the target function and set $K = h_1 = h_2, = \cdots = h_K$. We set $K = 6$ and use a single $\beta$ parameter. To avoid hyperparameter overfitting, we used the these hyperparameters for the SMM modules in *all* experiments. We use the exponential encoding to ensure positive weights. The weight parameters $z_i^{(k,j)}$ and the bias parameters were randomly initialized by samples from a Gaussian distribution with zero mean and unit variance truncated to $[-2, 2]$. We also used exponential encoding of $\beta$ and initialize $\ln \beta$ with $-1$.

We compared against isotonic regression (Iso) as implemented in the Scikit-learn library (Pedregosa et al., 2011) and XGBoost (XG, Chen & Guestrin, 2016). As initial experiments showed a tendency

of XG to overfit, we evaluated XG with and without early-stopping. We considered hierarchical lattice layers (HLL) as a state-of-the-art representative of lattice-based approaches using the well-documented implementation made available by the authors[1]. For a comparison of HLL with other lattice models we refer to Yanagisawa et al. (2022)). Furthermore, we applied LMNs using the implementation by Nolte et al..[2] For our new experiments, we adopted the basic architecture used in the *ChestXRay* experiments by Nolte et al. (2022) with two hidden layers and Lipschitz parameter one. The number of neurons in the hidden layers is determined by a width parameter. In each experiment, we considered two model sizes. The width parameter should be even, and we picked the width such that the model size (in degrees of freedom) of the small LMN$^s$ is smaller or equal to the size of the corresponding SMM. The larger LMN$^l$ used a width parameter increased by two compared to LMN$^s$. The resulting model sizes embrace the corresponding SMM model size, see next section and Table B.4 and Table C.9 in the appendix. In our experiments, the neural network models SMM, MM, HLL, and LMN were trained by the same unconstrained iterative gradient-based optimization procedure.

## 4.1 UNIVARIATE MODELLING

We considered three simple basic univariate functions on $[0, 1]$, the convex $f_{sq}(x) = x^2$, the concave $f_{sqrt}(x) = \sqrt{x}$, and the scaled and shifted logistic function $f_{sig} = (1 + \exp(-10(x - 1/2))^{-1}$; see also Yanagisawa et al. (2022) for experiments on $f_{sq}$ and $f_{sqrt}$. For each experimental setting, $T = 21$ independent trials were conducted. For each trial, the $N_{train} = 100$ training data points $\mathcal{D}_{train}$ were generated by randomly sampling inputs from the domain. Mean-free Gaussian noise with standard deviation $\sigma = 0.01$ was added to target outputs (i.e., the training data sets were typically not monotone as considered by Mikulincer & Reichman, 2022). The test data $\mathcal{D}_{test}$ were noise-free evaluations of $N_{test} = 1000$ evenly spaced inputs covering the input domain.

We compared SMM, MM, HLL, LMN as well as isotonic regression (Iso) and XGBoost (XG) with and without early-stopping. For $K = 6$, the MM and SMM modules have 72 and 73 trainable parameters, respectively. We matched the degrees of freedom and set the number of vertices in the HLM to 73; LMN$^s$ and LMN$^l$ had width parameters 6 and 8 resulting in 61 and 97 trainable parameters, respectively. We set the number of estimators in XGBoost to $n_{trees} = 73$ and $n_{trees} = 35$ (as the behavior was similar, we report only the results for $n_{trees} = 73$ in the following); for all other hyperparameters the default values were used. When using XGBoost with early-stopping, referred to as XG$_{val}$, we used 25 % of the training data for validation and set the number of early-stopping rounds to $\lfloor n_{trees}/10 \rfloor$. The isotonic regression baseline requires specifying the range of the target functions, and also HLL presumes a codomain of $[0, 1]$. This is useful prior information not available to the the other methods, in particular as some of the training labels may lie outside this range because of the added noise. We evaluated the methods by their mean-squared error (MSE). Details of the gradient-based optimization are given in Appendix A.

The test and training results of the experiments on the univariate functions are summarized in Table 1 and Table C.5, respectively. The distribution of the results is visualized in Figure C.3. In all experiments SMM gave the smallest median test error, and all differences between SMM and the other methods were statistically significant (paired two-sided Wilcoxon test, $p < 0.001$). The lower training errors of XG and Iso indicate overfitting. However, in our experimental setup, early-stopping in XG$_{val}$ did not improve the overall performance. The lattice layer performed better than XGBoost. SMM was statistically significantly better than HLL and both LMN variants;' the latter did not perform well in this experimental setup. Figure C.4 depicts the results of a random trial, showing the different ways the models extra- and interpolate.

Overall, SMM clearly outperformed MM. The variance of the MM learning processes was significantly higher, see Figure C.3. This can be attributed to the problem of silent neurons; the MM training got stuck in undesired local minima. When looking at the $3 \cdot 21 = 63$ trials on the univariate test functions, after training the maximum number of MM neurons at least once active over the test data set was as low as 5 out of 36; the mean number of active neurons was 2.8. On average 3.7 neurons in a network were active directly after initialization, that is, the training typically decreased the number

---

[1] https://ibm.github.io/pmlayer
[2] https://github.com/niklasnolte/MonotonicNetworks

of active neurons.[3] For SMM, we inspected the sum of the test predictions $\sum_{(x,y)\in\mathcal{D}_{\text{test}}} y_{\text{SMM}}(x)$ after training. We counted for how many neurons both partial derivatives of this sum w.r.t. the neuron's parameters were zero, which could happen for numerical reasons. This was rarely the case, on average 35 neurons were active after training using this notion of activity. Detailed results for MM and SMM are given in Table C.6 in the appendix.

*After these experiments,* we evaluated the robustness of the SMM results for different hyperparameters $\ln\beta \in \{-3,-2,-1,0,1\}$ and $K = h_k \in \{2,4,6,8\}$. The results are shown in Table C.7 in the appendix. Our default choice of $\beta = -1$ with $K = 6$ was suboptimal in all cases, although $\ln\beta \in \{-1,0\}$ gave overall good results. Decreasing $\beta$ further decreased the overall performance.

Table 1: Test errors on univariate (top) and mutivariate (bottom) tasks. A star indicates that the difference on the test data in comparison to SMM is statistically significant (paired two-sided Wilcoxon test, $p < 0.001$). The mean-squared error (MSE) values are multiplied by $10^3$.

|  | MM | SMM | XG | $XG_{\text{val}}$ | Iso | HLL | $LMN^s$ | $LMN^l$ |
|---|---|---|---|---|---|---|---|---|
| $f_{\text{sq}}$ | 0.10* | **0.01** | 0.14* | 0.18* | 0.04* | 0.04* | 0.37* | 0.09* |
| $f_{\text{sqrt}}$ | 0.32* | **0.02** | 0.14* | 0.20* | 0.06* | 0.06* | 0.28* | 0.27* |
| $f_{\text{sig}}$ | 0.22* | **0.01** | 0.13* | 0.17* | 0.04* | 0.04* | 0.25* | 0.26* |

|  | SMM | $XG^s$ | $XG^s_{\text{val}}$ | $XG^l$ | $XG^l_{\text{val}}$ | $HLL^s$ | $HLL^l$ | $LMN^s$ | $LMN^l$ |
|---|---|---|---|---|---|---|---|---|---|
| $d = 2$ | **0.00** | 0.23* | 0.26* | 0.23* | 0.26* | 0.03* | 0.03* | 0.07* | 0.03* |
| $d = 4$ | **0.01** | 0.66* | 0.76* | 0.66* | 0.76* | 0.03* | 0.08* | 0.29* | 0.06* |
| $d = 6$ | **0.02** | 0.74* | 0.82* | 0.74* | 0.82* | 0.10 | 0.13* | 0.07 | 0.07* |

## 4.2 MULTIVARIATE FUNCTIONS

We evaluated SMM, XG, HLL and LMN on multivariate monotone target functions. The original MM was dropped because of the previous results, Iso because the considered algorithm does not extend to multiple dimensions in a canonical way (the Scikit-learn implementation only supports univariate tasks). We considered three input dimensionalities $d \in \{2,4,6\}$. In each trial, we randomly constructed a function. Each function mapped a $[0,1]^d$ input to its polynomial features up to degree 2 and computed a weighted sum of these features, where for each function the weights were drawn independently uniformly from $[0,1]$ and then normalized by the sum of the weights. For example,

for $d = 2$ we had $(x_1,x_2)^{\text{T}} \mapsto (w_1 + w_2 x_1 + w_3 x_2 + w_4 x_1^2 + w_5 x_2^2 + w_6 x_1 x_2) \cdot \left(\sum_{i=1}^{6} w_i\right)^{-1}$

with $w_1,\ldots,w_6 \sim U(0,1)$. We uniformly sampled $N_{\text{train}} = 500$ and $N_{\text{test}} = 1000$ training and test inputs from $[0,1]^d$, and noise was added as above.

For $K = 6$, the dimensionalities result in 109, 181, and 253 learnable parameters for the SMM. The number of learnable parameters for HLL is given by the $L^d$ vertices in the lattice. In each trial, we considered two lattice sizes. For $HLL^s$, we set $L$ to 10, 3, and 2 for $d$ equal to 2, 4, and 6, respectively; for $HLL^l$ we increased $L$ to 11, 4, and 3, respectively. We also considered two LMN architectures. For both LMN and HLL the smaller network had fewer and the larger had more degrees of freedom than the corresponding SMM, see Table C.9 in the appendix. We ran XGBoost with $n_{\text{trees}} = 100$ ($XG^s$) and $n_{\text{trees}} = 200$ ($XG^l$), with and without early-stopping.

The test error results of $T = 21$ trials are summarized in Table 1. The corresponding training errors are shown in Table C.8 in the appendix. The boxplot of Figure C.5 in the appendix visualizes the results. The newly proposed SMM statistically significantly outperformed all other algorithms in all settings, except $HLL^s$ and $LMN^s$ for $d = 6$ where the lower errors reached by SMM are not significant. Using early stopping did not improve the XGBoost results in our setting, and doubling the number of trees did not have a considerable effect on training and test errors. We also measured the neural network training times for 1000 iterations, see Table C.9 in the Appendix C. $HLL^s$ was more than an order of magnitude slower than $LMN^s$ and the fastest method SMM.

---

[3]Before developing the SMM, we tried to solve the problem of silent neurons by improving the initialization, however, without success.

### 4.3 UCI PARTIAL MONOTONE FUNCTIONS

As a proof of concept, we considered modelling partial monotone functions on real-world data sets from the UCI benchmark repository (Dua & Graff, 2017). Details about the experiments are provided in Appendix B. We took all regression tasks and constraints from the first group of benchmark functions considered by Yanagisawa et al. (2022). The input dimensionality $d$ and number of constraints $|\mathcal{X}|$ were $d = 8$ and $|\mathcal{X}| = 3$ for the *Energy Efficiency* data (Tsanas & Xifara, 2012) (with two regression targets $Y_1$ and $Y_2$), $d = 6$ and $|\mathcal{X}| = 2$ for the *QSAR* data (Cassotti et al., 2015), and $d = 8$ and $|\mathcal{X}| = 1$ for *Concrete* (Yeh, 1998). We performed 5-fold cross-validation. From each fold available for training, 25 % were used as a validation data set for early-stopping and final model selection, giving a 60:20:20 split in accordance with Yanagisawa et al. (2022). In the partial monotone setting, HLL internally uses an auxiliary neural network. We used a network with a single hidden layer with 64 neurons, which gave better results than the larger default network. We considered SMM with unrestricted weights for the unconstrained inputs. We also added an auxiliary network. The $\text{SMM}_{64}$ model computes $a^{(k,j)}(\boldsymbol{x}) = \boldsymbol{w}^{(k,j)} \cdot \boldsymbol{x} + \Phi(\boldsymbol{x}_{\text{u}}) - b^{(k,j)}$, where $\Phi : \mathbb{R}^{d-|\mathcal{X}|} \to \mathbb{R}$ is a neural network with 64 hidden units processing the unconstrained inputs, see Appendix B for details. Similar to HLL, we incorporate the knowledge about the targets being in $[0, 1]$ by applying a standard sigmoid to the activation of the output neuron.

The mean cross-validation test error is shown in Table 2. $\text{SMM}_{64}$ performed best for one task, XG in the others. $\text{SMM}_{64}$ had the lowest CV test error of the neural network approaches on the two Energy tasks, and the larger LMN on QSAR and on Concrete.

### 4.4 COMPARISON WITH RECENTLY PUBLISHED RESULTS

The question arises how our approach compares to the results on larger real-world data sets presented by Nolte et al. (2022). Thanks to Nolte et al. who make their code for their experiments available,[4] we could evaluate SMM *exactly* as in their work. We employed the $\text{SMM}_{64}$ model already used in Section 4.3. As done by Nolte et al. (2022), we conducted only three trials, not enough to establish that the observed differences are statistically significant. Note that the evaluation procedure implemented by Nolte et al. (2022) assumes an oracle identifying the network with the lowest test error during training (i.e., the results in Table 3 are not unbiased estimates of generalization performance). It has to be stressed that the LMN results presented by Nolte et al. (2022) were produced using different network architectures and different hyperparameters of the learning algorithm for the different tasks. In contrast, we achieved our results using a single architecture which was not tuned for the tasks. We also used exactly the same number of training steps, we only adjusted the learning rates. For the *Heart Disease* task, we also provide the results when adding an additional sigmoid to the output and a slightly longer training time.

We added our experimental results to the values given in the Table 1 by Nolte et al. (2022), see our Table 3, which also contains results for certified monotonic neural networks (Liu et al., 2020) and counterexample-guided learning of monotonic neural networks (COMET, Sivaraman et al., 2020). SMM models gave better results in all of the benchmarks. For *BlogFeedback* we profited from the feature selection used by Nolte et al.. For Heart Disease, the architecture with the additional output sigmoid gave the best results (if we use the same number of training iterations the average result equals the 89.6 reported for LMN).

## 5 CONCLUSIONS

The smooth min-max (SMM) architecture, although only slightly different, improves on its ancestor MM both conceptually and in terms of accuracy in our empirical evaluation. An SMM module is a simple – and we would argue very elegant – way to ensure monotonicity. In light of our experimental results, many neural network approaches for modelling monotonic functions appear overly complex, both in terms of algorithmic description length and especially computational complexity. For example, lattice-based approaches suffer from the exponential increase in the number of trainable parameters with increasing dimensionality, and other approaches rely on solving SMT and MILP problems, which are typically NP-hard. The SMM is designed to be a module usable in a larger learning system

---

[4] https://github.com/niklasnolte/MonotonicNetworks

Table 2: Results on partial monotone UCI tasks, cross-validation error averaged over the MSE of 5 folds. The MSE is multiplied by 100. The dof columns give the numbers of trainable parameters, $n_{\text{trees}}$ the maximum number of estimators in XGBoost.

| | $\text{SMM}_{64}$ | | SMM | | XG | | HLL | | $\text{LMN}^{\text{s}}$ | | $\text{LMN}^{\text{l}}$ | |
| | $\mathcal{D}_{\text{test}}$ | dof | $\mathcal{D}_{\text{test}}$ | dof | $\mathcal{D}_{\text{test}}$ | $n_{\text{trees}}$ | $\mathcal{D}_{\text{test}}$ | dof | $\mathcal{D}_{\text{test}}$ | dof | $\mathcal{D}_{\text{test}}$ | dof |
|---|---|---|---|---|---|---|---|---|---|---|---|---|
| Energy $Y_1$ | **0.14** | 774 | 0.25 | 325 | 0.22 | 100 | 0.45 | 2139 | 0.27 | 727 | 0.22 | 841 |
| Energy $Y_2$ | 0.24 | 774 | 0.61 | 325 | **0.11** | 100 | 0.29 | 2139 | 0.44 | 727 | 0.34 | 841 |
| QSAR | 1.03 | 638 | 1.02 | 253 | **0.98** | 100 | 0.99 | 905 | 1.01 | 581 | 0.99 | 683 |
| Concrete | 1.78 | 902 | 1.79 | 325 | **1.71** | 100 | 4.59 | 707 | 2.20 | 841 | 1.71 | 963 |

Table 3: Comparison with results from Nolte et al. (2022) using the code from the authors and exactly the same setup (three trials, etc.), see caption of their Table 1. Accuracies and corresponding standard deviations are given in percent.

| Method | COMPAS $\uparrow\uparrow$ Test Acc | BlogFeedback $\downarrow\downarrow$ RMSE | LoanDefaulter $\uparrow\uparrow$ Test Acc | ChestXRay $\uparrow\uparrow$ Test Acc pretrained | $\uparrow\uparrow$ Test Acc end-to-end |
|---|---|---|---|---|---|
| Certified | $68.8 \pm 0.2$ | $0.158 \pm 0.001$ | $65.2 \pm 0.1$ | $2.3 \pm 0.2$ | $66.3 \pm 1.0$ |
| LMN | $69.3 \pm 0.1$ | $0.160 \pm 0.001$ | $65.44 \pm 0.03$ | $67.6 \pm 0.6$ | $70.0 \pm 1.4$ |
| LMN mini | | $0.155 \pm 0.001$ | $65.28 \pm 0.01$ | | |
| $\text{SMM}_{64}$ | **69.5 $\pm$ 0.1** | $0.192 \pm 0.002$ | $65.41 \pm 0.03$ | **67.9 $\pm$ 0.4** | **70.1 $\pm$ 1.2** |
| $\text{SMM}_{64}$ mini | | **0.154 $\pm$ 0.0004** | **65.47 $\pm$ 0.003** | | |

| Method | Heart Disease $\uparrow\uparrow$ Test Acc | Auto MPG $\downarrow\downarrow$ MSE |
|---|---|---|
| COMET | $86 \pm 3$ | $8.81 \pm 1.81$ |
| LMN | $89.6 \pm 1.9$ | $7.58 \pm 1.2$ |
| $\text{SMM}_{64}$ | $88.5 \pm 1.0$ [w/ sigmoid: **91.3 $\pm$ 1.89**] | **7.51 $\pm$ 1.6** |

that is trained end-to-end. From the methods considered in this study, MM, HLL, and LMN have this property, and we regard SMM as a drop-in replacement for those.

Which of the monotonic regression methods considered in this study results in a better generalization performance is of course task dependent. The different models have different inductive biases. All artificial benchmark functions considered in our experiments were smooth, matching the – rather general and highly relevant – application domain the SMM module was developed for. On a staircase function XGBoost would most likely outperform the neural networks. The monotonicity constraints of SMM act as a strong regularizer, and overfitting was no problem in our experiments. All SMM experiments were performed with a single hyperparameter setting for the architecture, indicating the robustness of the method. We regard the way SMM networks inter- and extrapolate (see Figure 1 and Figure C.4) as a big advantage over XG, HLL, and Iso for the type of scientific modelling tasks that motivated our work. LMNs share many of the desirable properties of SMMs. Imposing an upper bound on the Lipschitz constant, which is required for LMNs and can be optionally added to SMMs, can act as a regularizer and supports theoretical analysis of the neural network. However, a wrongly chosen bound can limit the approximation capabilities. Our experiments show that there are no reasons to prefer LMNs over SMMs because of generalization performance and efficiency. Because of the high accuracies obtained without architecture tuning, the more general asymptotic approximation results, and the simplicity of the min-max approach, we prefer SMMs over LMNs.

## REPRODUCIBILITY STATEMENT

All experiments, plots, tables, and statistics can be reproduced using the submitted source code in the supplmentary material. The code for HLL and LMN models is kindly made available by the original authors from `https://ibm.github.io/pmlayer` and `https://github.com/niklasnolte/MonotonicNetworks`. The experiments by Nolte et al. (2022) use the code from `https://github.com/niklasnolte/monotonic_tests`.

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

## A  GRADIENT-BASED OPTIMIZATION

The neural network models SMM, MM, and HLL were fitted by unconstrained iterative gradient-based optimization of the mean-squared error (MSE) on the training data. We used the Rprop optimization algorithm (Riedmiller & Braun, 1993). On the fully monotone benchmark functions, we did not have a validation data set for stopping the training. Instead, we monitored the *training progress over a training strip of length* $k$ defined by Prechelt (2012) as $P_k(t) = 10^3 \cdot \left( \frac{\sum_{t'=t-k+1}^{t} E_{\text{train}}(t')}{k \cdot \min_{t'=t-k+1}^{t} E_{\text{train}}(t')} \right)$ for $t \geq k$. Here $t$ denotes the current iteration (epoch) and $E_{\text{train}}(t')$ the MSE on training data at iteration $t'$. Training is stopped as soon the progress falls below a certain threshold $\tau$. We used $k = 5$ and $\tau = 10^{-3}$. This is a very conservative setting which worked well for HLL and was then adopted for all algorithms.

## B  DETAILS ON UCI EXPERIMENTS

The experiments on partial monotone functions were inspired by Yanagisawa et al. (2022). As briefly discussed in Section 4, a fair comparison on complex partial monotone real-world tasks is challenging. There is the risk that the performance on the unconstrained features overshadows the processing of the constraint features. Therefore, we did not consider the second group of UCI tasks from the study by Yanagisawa et al. (2022), because the fraction of constrained features in these problems is too low – and we would argue that the low number of constrained features already is an issue for the problems in the first group when evaluating monotone modelling. We selected all regression tasks from the first group, see the overview in Table 2. We used the same constraints, see Table 2, and normalization to $[0, 1]$ of inputs and targets as Yanagisawa et al. (2022).

Furthermore, architecture and hyperparameter choices become more important in the UCI experiments compared to the experiments on the comparatively simple benchmark functions. For partial monotone tasks, the HLL requires an auxiliary neural network. The default network did not give good results in initial experiments, so we replaced it by a network with a single hidden layer with 64 neurons, which performed considerably better. The lattice sizes of the constrained input features were set to $k = 3$.

For a fair comparison, we also added an auxiliary network with 64 neurons to the SMM module. For complex real-world tasks, an isolated SMM module with a single layer of adaptive weights – despite the asymptotic approximation results – is not likely to be the right architecture. Thus, we considered SMM modules with a single neural network $\Phi : \mathbb{R}^{d-|\mathcal{X}|} \to \mathbb{R}^{d}$ with one hidden layer and compute $a^{(k,j)}(\boldsymbol{x}) = \boldsymbol{w}^{(k,j)} \cdot \boldsymbol{x} + \Phi(\boldsymbol{x}_{\text{u}}) - b^{(k,j)}$, where $d$ is the input dimensionality, $\boldsymbol{x}_{\text{u}}$ are the unconstrained inputs, $|\mathcal{X}|$ is the number of constrained variables, and $\forall m \in \mathcal{X} : w_m^{(k,j)} \geq 0$, see end of Section 3. We set the number of hidden neurons of $\Phi$ to 64, so that degrees of freedom are similar to the HLL employed in our experiments. Also similar to HLL, we incorporate the knowledge about the targets being in $[0, 1]$ by applying a standard sigmoid $\sigma$ to the activation of the output neuron. The resulting architecture, which we refer to as $\text{SMM}_{64}$, can alternatively be written as as a residual block computing $\sigma(y(\boldsymbol{x}) + \Phi(\boldsymbol{x}_{\text{u}}))$, where $y(\boldsymbol{x})$ is the standard SMM. This may be the simplest way to augment the SMM.

Table B.4: UCI regression data sets and constraints as considered by Yanagisawa et al. (2022). The input dimensionality is denoted by $d$, the number of data points by $n$. The last five columns give the number of trainable parameters of the models used in the experiments; SMM and $\text{SMM}_{64}$ denote the smooth min-max network without and with $\Phi$.

|          | $d$ | $n$  | monotone features | no. parameters | | | | |
|----------|-----|------|-------------------|------|--------------|-----|------|------|
|          |     |      |                   | SMM  | $\text{SMM}_{64}$ | HLL | LMN$^{\text{s}}$ | LMN$^{\text{l}}$ |
| Energy   | 8   | 768  | X3, X5, X7        | 325  | 744          | 2139 | 727  | 841  |
| QSAR     | 6   | 908  | MLOGP, SM1_Dz(Z)  | 253  | 638          | 905  | 581  | 683  |
| Concrete | 8   | 1030 | Water             | 325  | 902          | 707  | 841  | 963  |

We performed 5-fold cross-validation to evaluate the methods. Each data fold available for training was again split to get a validation data set, giving a 60:20:20 spit into training, validation, and test

data as considered by Yanagisawa et al. (2022). We monitored the MSE on the validation data during training and stored the model with the smallest validation loss. If the validation error did not decrease for 100 epochs, the training was stopped.

## C  ADDITIONAL RESULTS

Table C.5: Training errors on univariate tasks. The mean-squared error (MSE) values are multiplied by $10^3$.

|  | MM | SMM | XG | $XG_{val}$ | Iso | HLL | $LMN^s$ | $LMN^l$ |
|---|---|---|---|---|---|---|---|---|
| $f_{sq}$ | 0.17 | 0.10 | 0.05 | 0.11 | 0.03 | 0.03 | 0.42 | 0.14 |
| $f_{sqrt}$ | 0.35 | 0.09 | 0.04 | 0.10 | 0.03 | 0.03 | 0.31 | 0.25 |
| $f_{sig}$ | 0.27 | 0.10 | 0.05 | 0.11 | 0.04 | 0.04 | 0.36 | 0.43 |

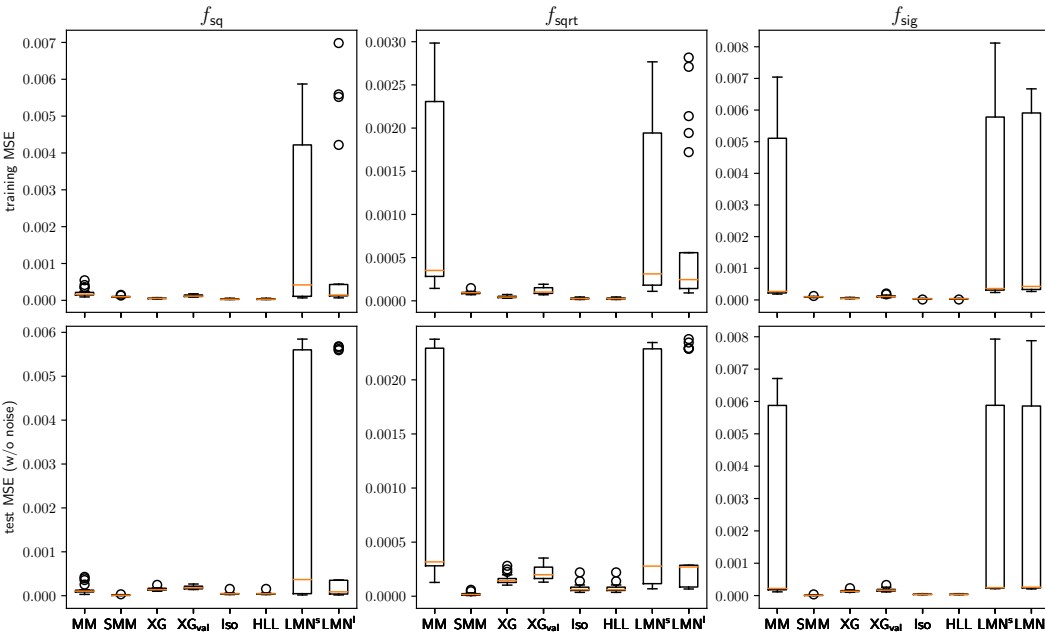

Figure C.3: Results on univariate functions based on $T = 21$ trials. Depicted are the median, first and third quartile of the MSE (without clipping the outputs to the target function codomain); the whiskers extend the box by $1^{1}/_{2}$ the inter-quartile range, dots are outliers. Training errors are shown in the top, test errors in the bottom row.

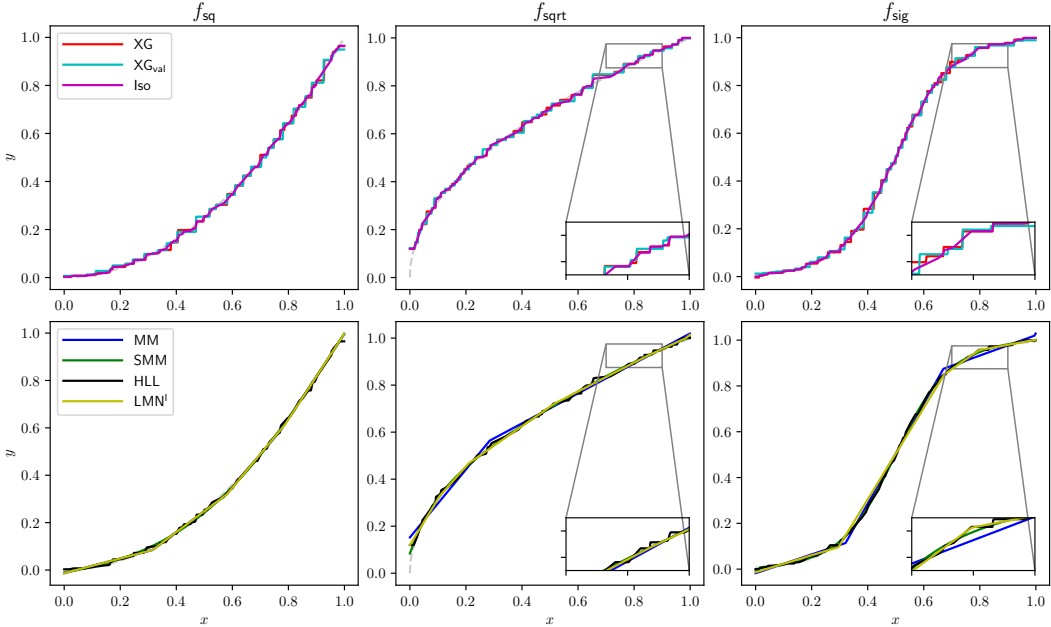

Figure C.4: Function approximation results of a single trial (outputs not clipped) for each of the three univariate functions. The top row shows the non-neural, the bottom row the neural methods.

Table C.6: Active neurons on univariate tasks when evaluated on the test sets. For MM, a neuron was not active in a trial if it never contributed to an output when the network was evaluated on the test data. For SMM, a neuron was regarded as not active in a trial if the partial derivatives of the sum of the predictions on the test set w.r.t the parameters of the neuron were all zero. For MM, we report the number of active neurons before and after training.

| | | MM | | | | | | SMM | |
| | | initial | | | final | | | | |
| | min | mean | max | min | mean | max | min | mean | max |
|---|---|---|---|---|---|---|---|---|---|
| $f_{sq}$ | 1 | 3.4 | 6 | 2 | 3.4 | 5 | 34 | 35.7 | 36 |
| $f_{sqrt}$ | 2 | 3.9 | 7 | 1 | 2.1 | 4 | 29 | 33.8 | 36 |
| $f_{sig}$ | 1 | 3.7 | 7 | 2 | 3.0 | 5 | 34 | 35.6 | 36 |
| overall | 1 | 3.7 | 7 | 1 | 2.8 | 5 | 29 | 35.0 | 36 |

Table C.7: Test errors on univariate tasks for SMM different choices of $K$ and initial $\beta$. Shown are medians over 11 trails. The mean-squared error (MSE) values are multiplied by $10^3$.

| $K$ | -3 | -2 | -1 | 0 | 1 |
|---|---|---|---|---|---|
| | | | $\ln \beta$ | | |
| | | | $f_{\mathrm{sq}}$ | | |
| 2 | 0.0114 | **0.0055** | 0.0111 | 0.0121 | 0.0087 |
| 4 | 0.0069 | 0.0068 | 0.0064 | 0.0077 | 0.0077 |
| 6 | 0.0064 | 0.0062 | 0.0072 | 0.0065 | 0.0062 |
| 8 | 0.0070 | 0.0079 | 0.0080 | 0.0068 | 0.0073 |
| | | | $f_{\mathrm{sqrt}}$ | | |
| 2 | 0.1030 | 0.0752 | 0.0756 | 0.0712 | 0.0700 |
| 4 | 2.2955 | 2.2624 | 0.0157 | 0.0156 | 0.0184 |
| 6 | 2.2960 | 2.2882 | 0.0220 | 0.0164 | 0.0180 |
| 8 | 2.2976 | 0.0297 | 0.0177 | **0.0123** | 0.0191 |
| | | | $f_{\mathrm{sig}}$ | | |
| 2 | 7.8617 | 0.0154 | 0.0058 | 0.0056 | 0.0055 |
| 4 | 7.8544 | 0.0096 | **0.0051** | 0.0076 | 0.0123 |
| 6 | 0.1012 | 0.0062 | 0.0052 | 0.0084 | 0.0113 |
| 8 | 7.8559 | 0.0058 | 0.0054 | 0.0080 | 0.0119 |

Table C.8: Multivariate tasks, training error. The mean-squared error (MSE) values are multiplied by $10^3$.

| | SMM | $\mathrm{XG}^{\mathrm{s}}$ | $\mathrm{XG}^{\mathrm{s}}_{\mathrm{val}}$ | $\mathrm{XG}^{\mathrm{l}}$ | $\mathrm{XG}^{\mathrm{l}}_{\mathrm{val}}$ | $\mathrm{HLL}^{\mathrm{s}}$ | $\mathrm{HLL}^{\mathrm{l}}$ | $\mathrm{LMN}^{\mathrm{s}}$ | $\mathrm{LMN}^{\mathrm{l}}$ |
|---|---|---|---|---|---|---|---|---|---|
| $d = 2$ | 0.10 | 0.14 | 0.19 | 0.14 | 0.19 | 0.08 | 0.07 | 0.16 | 0.12 |
| $d = 4$ | 0.10 | 0.19 | 0.33 | 0.19 | 0.33 | 0.09 | 0.06 | 0.27 | 0.15 |
| $d = 6$ | 0.09 | 0.13 | 0.30 | 0.13 | 0.30 | 0.14 | 0.03 | 0.15 | 0.14 |

Table C.9: Multivariate tasks, degrees of freedom of the neural networks and accumulated training times (on a Apple M1 Pro) in seconds for conducting 21 trials with 1000 training steps each.

| | SMM | | $\mathrm{HLL}^{\mathrm{s}}$ | | $\mathrm{HLL}^{\mathrm{l}}$ | | $\mathrm{LMN}^{\mathrm{s}}$ | | $\mathrm{LMN}^{\mathrm{l}}$ | |
|---|---|---|---|---|---|---|---|---|---|---|
| | time (s) | dof | time (s) | dof | time (s) | dof | | | | |
| $d = 2$ | 11.54 | 109 | 403.87 | 100 | 498.71 | 121 | 12.01 | 105 | 12.48 | 151 |
| $d = 4$ | 11.47 | 181 | 358.83 | 81 | 1351.90 | 256 | 12.50 | 171 | 12.75 | 229 |
| $d = 6$ | 11.43 | 253 | 301.45 | 64 | 8215.19 | 729 | 12.81 | 253 | 13.27 | 323 |

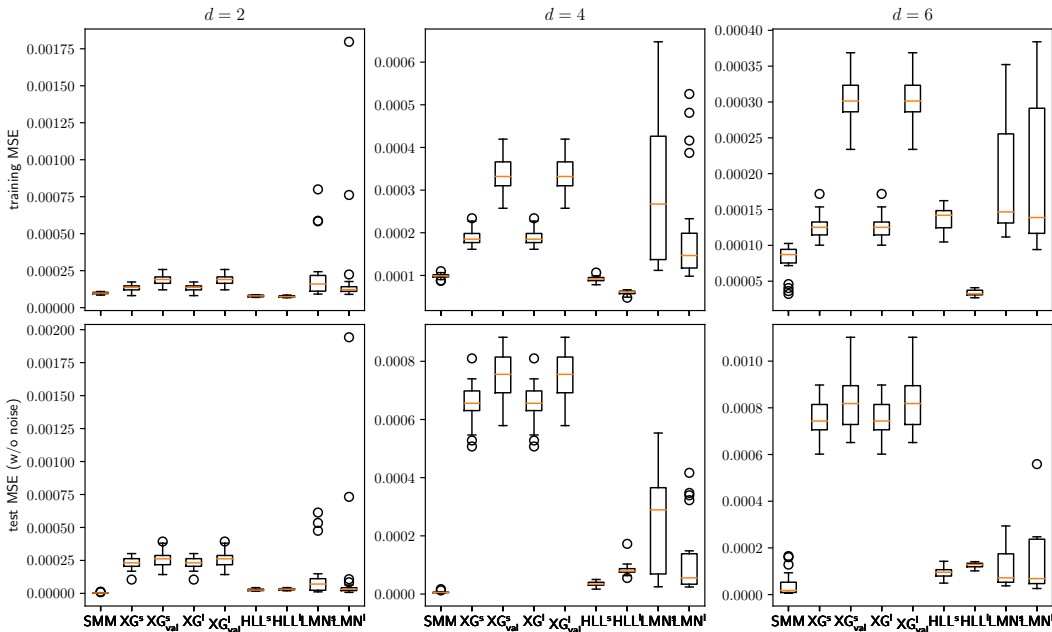

Figure C.5: Results on multivariate functions based on $T = 21$ trials. Depicted are the median, first and third quartile of the MSE; the whiskers extend the box by $1^1/2$ the inter-quartile range, dots are outliers. Early-stopping reduced the XGBoost training accuracy, but did not lead to an improvement on the test data.

