# OpenReview forum: "Smooth Min-Max Monotonic Networks"
_ICLR.cc/2024/Conference — Submitted to ICLR 2024_

### Official Review · Reviewer_RbCP · 2023-10-29

**Soundness:** 4 excellent
**Presentation:** 4 excellent
**Contribution:** 3 good
**Rating:** 6
**Confidence:** 4

**Summary:**

This paper addresses an intriguing aspect of machine learning: monotonic modeling, focusing specifically on the min-max architecture. The authors thoroughly summarize various techniques and identify a key issue for min-max architecture known as the "silent neuron" problem. In response, they propose a smooth variant and develop what they term the SMM architecture. This new architecture demonstrates strong experimental results.

**Strengths:**

This SMM architecture is not only innovative but also well-motivated solution by transitioning from the conventional hard min-max to a LogSumExp-based approach. Furthermore, the paper establishes theoretical guarantees about model's approximation property when the parameter $\beta$ is sufficiently large.

The experimental results are another major strength of this work. The authors demonstrate the effectiveness of the smooth min-max (SMM) architecture, thereby confirming both the practicality and the potential of their approach.

**Weaknesses:**

One significant concern lies in the treatment of $\beta$ as a learnable parameter. The authors' exploration of this parameter is fascinating, particularly in light of Corollary 1's suggestion that a lower bound on fitting error is inherently linked to the value of $\beta$. This implies that a $\beta$ not sufficiently large would fail to approximate certain functions. Conversely, an excessively large $\beta$ might impact the training dynamics adversely, as some nearly silent neurons may remain untrained.

While the authors utilize trainable $\beta$ in experiments, the paper could benefit from a deeper exploration of $\beta$'s behavior during training, such as its trajectory and its relationship with loss changes. Reporting the final values of $\beta$ after training would also have provided valuable insights.

**Questions:**

The observation that test errors can vary significantly with different initial $\beta$ values raises an important question.
Does it suggest that the optimization process may not fully converge or that $\beta$ plays a more complex role in the model's training dynamics than currently understood?

---

> ### Author Response · Authors · 2023-11-15
> **Initial reply to review**
>
> Thank you for your feedback.
>
> Corollary 1 is an existence proof, showing that for a certain beta value an SMM with a particular approximation accuracy exists. This network – and beta parameter – may not be the “best” one for the given task, there may be SMNs that have fewer parameters, that have smaller weights, are Lipschitz with a smaller constant, are faster to learn, etc. with the same or better approximation accuracy. Thus, Corollary 1 is not strong enough to establish a link between fitting error and the value of beta for practical experiments as considered in the paper.
>
> Thank you for the question regarding beta. In general, too extreme beta values may cause numerical trouble, which leads to vanishing gradients. If beta is too large, then the SMN gets stuck in undesired local minima as the original MM. To show this, we reran the experiments underlying Table C.4 with ln(beta)=2. This reduced the mean number of active SMM neurons from 35 to 29.3 (means for $f_\text{sq}$, $f_\text{sqrt}$, and $f_\text{sig}$ were 31.81, 30.71, 25.42, respectively).
>
> We are happy to add an extended analysis to the paper.

---

> > ### Comment · Reviewer_RbCP · 2023-11-23
> > **Response to author's response**
> >
> > Indeed, Corollary 1 is an existence proof. However, by looking into the proof, I only see why an approximation could exist when $\beta$ is sufficiently large. I am not sure what is the “best” $\beta$. It seems network with smaller $\beta$ is always representable by another network with larger $\beta$. In theory, the merit of small $\beta$ only lies in avoiding vanishing gradient. Yet the link between that with local minima is not clear. I think the role of $\beta$ is still not totally clear.
> >
> > Despite this uncertainty, I am inclined to keep my rating of weakly accept for the innovation of LogSumExp-based approach.

---

> > > ### Author Response · Authors · 2023-11-23
> > > **Role of beta**
> > >
> > > Thank you for your reply!
> > > Perhaps the following makes the role of beta clearer. You can think about beta in a different way.
> > > From eq. (4) we see that $\beta$ occurs twice: It linearly rescales the input and its inverse rescales the output.
> > > The inputs in (7) are affine linear functions. The input rescaling by beta can be replaced by simply changing the weights $w$ and $b$ (multiplying by beta) of the linear functions.
> > > The output scaling by $\frac{1}{\beta}$ in (7) meets in eq. (8) the input scaling by $\beta$, that is, all betas "in the middle" cancel.
> > > What remains is that the final output is simply rescaled by $\beta$, which can dropped without changing the monotonicity of the network or counterbalanced by adding an additional positive weight $w_{\text{out}}\in\mathbb{R}^+$ rescaling the output. This should be added to the paper.

---

### Official Review · Reviewer_odc5 · 2023-10-31

**Soundness:** 3 good
**Presentation:** 3 good
**Contribution:** 2 fair
**Rating:** 3
**Confidence:** 3

**Summary:**

The paper studies the training and empirical performance of neural networks and non-neural approaches that ensures monotonicity with respect to input parameters. The authors propose a new network module architecture based on min-max (MM) architecture [Sill (1997)] which aims to tackle the problem of silent neurons and non-smoothness properties by applying a LogSumExp function to the max/min function. The authors support their claims by providing empirical evidence on toy examples and on practical data sets.

**Strengths:**

1) This paper is well-written and is easy to follow. The authors presented their ideas and results clearly.
2) The proposed SMM architecture is simple and seems to be an intuitive way to ensure monotonicity through smoothening.
3) The authors did extensive comparisons of their proposed SMM against other models which aim to ensure monotonicity, and aided readers to understand the potential advantages of SMM over comparable models.

**Weaknesses:**

1) I am not entirely sure about the novelty of this idea of smoothening non-smooth neurons to address the problem of vanishing gradients or silent neurons in the context of monotonic networks. The main idea of this work of using LogSumExp to act as a smooth approximation while preserving monotonicity does not seem too non-trivial due to its popularity in statistic modelling. However, I am not familiar with the line of work with monotone networks thus I will defer this discussion to other reviewers.
2) While the empirical comparisons are sufficient, they do not provide evidence (especially after accounting the error bars) to suggest that SSM has significant advantage over existing approaches. It is then unclear why practitioners should prefer SSMs over LMNs or XGBoost.

**Questions:**

1) How should the scaling factor $\beta$ chosen in practice? My understanding is that tuning it to ensure that the output network is monotone is not trivial and requires retraining the entire network.

---

> ### Author Response · Authors · 2023-11-15
> **Initial reply to review**
>
> Thank you for your feedback.
>
> **Question 1:** "My understanding is that tuning it to ensure that the output network is monotone is not trivial and requires retraining the entire network.": *The network is monotone for all choices of beta.*
>
> Tuning beta is not an issue in practice. Note that beta is adapted during training. All experiments (except those in reported in Table C.7) were conducted with the same "default" initialization ln(beta)=-1. This shows the robustness of our method. We suggest to simply use the default initialization (much too large or too small initial beta values may impact the performance, see also reply to reviewer RbCP).
>
> **Weakness 1:** To the best of our knowledge, our idea is new in the context of monotonic neural networks. This is indeed remarkable, given the many papers building on Sill’s work coming up with complicated alternatives.
>
> **Weakness 2:** The results in Table 1 show statistically significant better performance of SMM compared to all other methods. The results in Table 1 may be regarded as the most relevant ones, because the experimental setup is very controlled and focussed on the monotonicity aspect.
>
> We are very sorry that we failed to make it clear in our text why one should prefer SSMs over LMNs or XGBoost in practice.
>
> XGBoost vs SMM: XGBoost models are piecewise constant. If you want a smooth model (e.g., in a natural science application), then you should prefer SMM or LMN over XGBoost. If you have a larger (deep) learning system and you need a monotonic module and would like to train end-to-end, then XGBoost is not a good option and you should pick a neural network such as LMN or SMM. If you do not have these constraints and tabular data, then XGBoost is a choice that is difficult to beat by any neural network approach - except perhaps SMMs as shown in Table 1.
>
> SMM vs LMN: “LMNs share many of the desirable properties of SMMs. Imposing an upper bound on the Lipschitz constant, which is required for LMNs and can be optionally added to SMMs, can act as a regularizer and supports theoretical analysis of the neural network. However, a wrongly chosen bound can limit the approximation capabilities. Our experiments show that there are no reasons to prefer LMNs over SMMs because of generalization performance and efficiency. Because of the high accuracies obtained without architecture tuning, the more general asymptotic approximation results, and the simplicity of the min-max approach, we prefer SMMs over LMNs.”
>
> Perhaps this was too modest, so we would like to add: Over all experimental results in our study, SMM performs (most often significantly) better than LMN. While SMMs are very robust w.r.t. to hyperparameter choices (as shown in the study), the Lipschitz bound to be chosen for LMNs might be a problem in practice. We see no reason to prefer LMN over SMM.

---

### Official Review · Reviewer_2p1V · 2023-11-04

**Soundness:** 3 good
**Presentation:** 3 good
**Contribution:** 2 fair
**Rating:** 3
**Confidence:** 2

**Summary:**

The authors propose modification to min-max networks by replacing max and min by appropriate log sum exp functions.

This is done to improve the learning signal.

Some theoretical/empirical analysis is provided.

**Strengths:**

The paper is very clear, I could understand most of it in first reading.

The authors consider an important problem: sometimes "worse" models can be empirically better as it is easier to optimise.

**Weaknesses:**

Are there different types of relaxation of min/max that can be used?

I think the results of type Thm 1 are not very meaningful as the network size can increase very quickly when epsilon decreases.

The empirical results are not very strong. Is e.g. ChestXRay statistically significant? The differences in Table 3 look mostly statistically insignificant.

**Questions:**

See above.

---

> ### Author Response · Authors · 2023-11-15
> **Initial reply to review**
>
> Thank you for your feedback.
>
> **I.** There are several smooth (approximations of the) maximum and minimum functions (e.g., computing the p-norm for large p). However, we need minimum and maximum relaxations which
> * are non-decreasing,
> * are smooth,
> * need to work for positive and negative arguments,
> * have a bounded approximation error which can be controlled (important for Corollary 1), and
> * can be computed efficiently without causing too much numerical trouble.
>
> The LogSumExp is the best choice we know fulfilling all these properties. If there was a better choice, this would make our idea work even better.
>
> **II.**  Theorem 1 and Corollary 1 have the standard form of a universal approximation theorem. If one wants universal approximation capabilities, there are functions for which the construction used in the proofs will require many neurons to achieve a certain accuracy. This is inevitable. Still, we regard these results as meaningful because they show a fundamental theoretical property of the model, an asymptotic property such as unbiasedness and consistency of learning methods and convergence to an optimum of a gradient-based optimizer. From a statistical learning theory perspective, universal approximation theorems are important, for example, to understand which learning problems are realizable given the neural network hypothesis class. In practice we would argue that the properties – even if asymptotic – can help to choose one algorithm over another.
>
> Studying the theoretical properties of machine learning algorithms is an important area of machine learning research. One can of course argue about the practical relevance of each of the properties listed above, however, one should appreciate that there is a large community who is interested in studying these properties ( https://en.wikipedia.org/wiki/Universal_approximation_theorem ).
>
> **III.** The results in Table 3 are not statistically significant. The motivation for Table 3 was to reproduce *exactly* the experimental setting from Niklas Nolte et al. published at ICLR last year so that the results can be compared directly. The experimental setup is not ideal: Only *three* trials, early-stopping on the test set, too complex tasks where monotonicity only plays a minor role, varying hyperparameters for the LMNs. The results show that you can get at least as good (we would argue better) results with SMMs than LMNs if you use exactly the evaluation protocol as in the LMN paper from ICLR last year.
>
> For a statistical comparison, the more important results are those in Table 1, which come with proper statistical significance testing. Here SMMs show significantly better results.

---

### Author Response · Authors · 2023-11-21

Hello,

We would be happy to engage in a discussion with the reviewers.
Did our replies answer all their questions?
There were some misunderstandings (e.g., regarding the dependency of the monotonicity on beta) - are these resolved?

We think that our submission presents an effective, elegant and theoretical sound solution to an important neural network learning task.
The conceptual arguments are supported by experiments, which are more extensive than in the related literature.
However, our research does not seem to be appreciated by the reviewers and we would like to better understand why.
We would be happy to provide additional information.

Thank you for your consideration!

---

### Meta-Review · Area_Chair_3BNL · 2024-01-12

**Metareview:**

This paper considers so-called monotonic modeling, using a min-max architecture, it identifies the "silent neuron" problem, and it proposes a smooth variant and develops a ned architecture.  There were three reviews, none of which were very positive.  One, which was more detailed, was weak accept; and two, which were less detailed, were reject.  The architecture and empirical results have some merit.  The lower reviewers had concerns that at a minimum could be considered to help improve the presentation of the paper.  The higher reviewer had several questions about the role of a parameter during the training process, including side-effects of it, it's connection with loss changes, etc.  These were partially addressed in a response, but a more detailed evaluation would also make the paper stronger.

**Justification For Why Not Higher Score:**

There were three reviews, none of which were very positive.  One, which was more detailed, was weak accept; and two, which were less detailed, were reject.

**Justification For Why Not Lower Score:**

It is low.

---

### Decision · Program_Chairs · 2024-01-16

Reject